# Simulation of Silicon Surface Barrier Detector with PN Junction Guard Rings to Improve the Breakdown Voltage

**DOI:** 10.3390/mi13111811

**Published:** 2022-10-23

**Authors:** Bolong Wang, Rui Jia, Xing Li, Ke Tao, Wei Luo, Longjie Wang, Jiawang Chen

**Affiliations:** 1Institute of Microelectronics, Chinese Academy of Sciences, Beijing 100029, China; 2University of Chinese Academy of Sciences, Beijing 100029, China

**Keywords:** surface barrier detector, guard rings, device simulations, breakdown voltage

## Abstract

Silicon surface barrier detectors (SSBDs) are normally used to detect high-energy particles due to their excellent properties. For better charge collection efficiency (CCE), the SSBD device should be operated at higher reverse voltages, but this can lead to device breakdown. Therefore, we used a PN junction as a guard ring to increase the breakdown voltage of the SSBD. The structures of two SSBD devices are drawn and simulated in this work. Compared with a conventional SSBD (c-SSBD), the use of a PN junction as a guard ring for an SSBD (Hybrid-SSBD) achieves higher breakdown voltages, of over 1500 V under reverse bias. This means that Hybrid-SSBD devices can operate at higher reverse voltages for better charge collection efficiency (CCE) to detect high-energy particles. Then, we simulated the different structure parameters of the Hybrid-SSBD guard rings. Among them, the doping depth and gap width of the guard ring (between the innermost guard ring and the active area) have a greater impact on the breakdown voltage. Finally, for Hybrid-SSBD devices, the optimal characteristics of the guard ring were 1 × 10^19^ cm^−3^ doping concentration, 1 μm doping depth, and innermost guard ring width and gap width of 5 μm and 3 μm, respectively.

## 1. Introduction

In early high-energy physics experiments, silicon surface barrier detectors (SSBDs) were used to detect the high-energy particles due to their relative ease of fabrication, room temperature operation, and good long−term stability [1,2,3,4]. The energy resolution of an SSBD device is closely related to charge collection efficiency and leakage current. To reduce the leakage current, many works to have attempted to modify the metal–semiconductor interface and enhance the Schottky barrier height [5,6,7]. For excellent detection efficiency, the detectors should operate at a high electric field or reduce the thickness of substrate. However, thinner substrates have higher process requirements, so increasing the electric field is an effective way to improve the charge collection efficiency. Thus, the detectors should be designed for higher breakdown voltage on high resistivity silicon [8]. Schottky diodes have a low breakdown voltage due to the edge effects caused by discontinuities in the metal electrodes. Currently, there is much ongoing research into designing reasonable edge termination to relieve the effect of electrical field crowding, such as field plates, guard rings, and junction termination extension [8,9,10]. Among these, guard rings, as a means of edge termination, can effectively improve the breakdown voltage of power devices. Typically, the guard ring should surround the active area, to establish a gradual potential drop between the active area with ground potential and the cutting edge with backside potential. This avoids the risk of breakdown induced by the high electric field at the edge of the device [11]. In recent research on guard rings, overlapping metal structures, implanted edge termination, and diffusion guard rings have been proposed to prevent discontinuities in metal electrode that preclude generating the high electric field [12,13,14,15]. SSBD devices are usually formed directly by metal deposition and the preparation of the guard ring is rarely considered. However, SSBD detectors should work on the higher reverse bias voltage to obtain a good charge collection efficiency. Currently, SSBDs have a low breakdown voltage, less than 300 V, so the guard rings should be added to the device to increase the breakdown voltage. Different structures for guard rings have been reported in power devices; however, there has been no such report on the simulation of SSBDs adopting a PN junction guard ring for edge termination.

Technology computer−assisted design (TCAD) can be used to design the device structure and to solve the partial differential equations to extract physical parameters such as charge carrier motion, the distribution of the electric field, and the electric potential. This simulator can be used to explore different device structure designs, and the influence of the parameters can also be simulated by using this simulator. In this paper, a hybrid structure of SSBD detector, which uses a Schottky diode as the active area and guard ring adopting a PN junction as the edge termination, is proposed to increase the breakdown voltage. We compared the breakdown voltage and analyzed the performance of conventional SSBDs, which use Schottky diodes as the active area and guard rings. For the hybrid SSBD, the effect of various structure parameters on the electrical performance, such as guard ring doping depth, guard ring doping, innermost guard width, and innermost guard ring−to−active area gap, were investigated.

## 2. Device Structure and Simulation Method

In order to simplify the simulation process and improve the calculation efficiency, a semi−infinite and symmetric device structure was used in this study, as shown in Figure 1. For the conventional SSBD (c-SSBD), both the guard rings and the active area are formed by Schottky diodes. For the hybrid−SSBD structure, in Figure 2, the active area at the central place of the device is formed by a Schottky contact, but the guard rings around the active area are formed by p+ ion implantation or p+ diffusion. The two devices are based on the high−resistance n−type silicon substrate; the doping concentration is lower than 1 × 10^12^ cm^−3^ and the thickness is 525 μm. Both have an active area width of 250 μm, and the number of guard rings is fixed at 7. In this paper, the innermost guard ring width is varied from 2 to 12 μm, and the distance between it and the active area varied from 3 to 10 μm. The doping concentration and depth of the p+ region varied in the range of 1 × 10^17^ to 1 × 10^21^ cm^−3^ and 0.1 to 2 μm, respectively.

The simulations were performed using 2−D TCAD device simulators, which is a finite element semiconductor simulation capable of simulating the processing and device performance in some physical situations [16,17,18]. In the simulation, the high−field saturation model and mobility degradation at interfaces were used. For recombination, doping− and temperature−dependent Shockley–Read–Hall and Auger models were included. For the high−resistance silicon substrate, the device breakdown was driven by the electric field, and avalanche breakdown would appear in the device edge or junction edge. This can be described by Equation (1); in this formula, *G* is the generation rate of the electron–hole pair caused by the avalanche generation, *α_n_* and *α_p_* are the ionization coefficients, and ***J***_***n***_ and ***J***_***p***_ are the absolute value of electron and hole current density vectors.
(1)$$G=\frac{1}{q}\left(\alpha_{n}J_{n}+\alpha_{p}J_{p}\right)$$

Several avalanche models can be activated to simulate the free carrier generation mechanism leading to avalanche breakdown. The avalanche model suggested by Okuto and Crowell was used to evaluate the breakdown behavior of the two kinds of structure [19].

## 3. Results and Discussion

### 3.1. c-SSBD and Hybrid-SSBD Performance Compared

Figure 3 displays the results of two structural simulations of the I–V curves. The 1500 V reverse sweep voltage was applied to both two devices while guard rings were kept floating and the current in the active area was measured. It was observed that there was a significant difference in the breakdown voltage of the two devices. For c-SSBD device, the leakage current reached saturation as the reverse voltage increased, and then the device broke down with a rapid increase in leakage current as the reverse bias voltage continued to increase (breakdown voltage ~231 V). However, the breakdown voltage of the Hybrid-SSBD device was higher than ~1450 V and the leakage current was lower than that of the SSBD device. The simulated I–V curves suggest that using PN junction guard rings could effectively prevent the breakdown of the Schottky barrier detector at higher reverse voltages.

As shown in Figure 4, the impact ionization rates of the two devices were used to analyze the location of the avalanche breakdown in the structure. In the figures, different colors correspond to different impact ionization rate intensities; the red areas appear at the edges of the active area and the guard rings, indicating that stronger impact ionization rates occurred in these areas. When comparing the two simulation results, there was significant different between the active area and the guard rings. The c-SSBD device had a higher impact ionization rate than the Hybrid-SSBD device at the edge of active area, which implied that, under the same bias voltage, c-SSBD devices broke down more easily.

We further analyzed the electric field distribution of the two devices, as shown in the Figure 5. It can be seen that the electric field distribution of the c-SSBD device is concentrated at the edge of the active area, which can easily cause breakdown. However, using the PN junction as the edge termination (Hybrid-SSBD) flattens the electric field across the device edge. In Figure 6, the electric field distribution of the two devices was extracted, and the uniform electric field distribution in the guard rings region is beneficial, improving the breakdown voltage of the Hybrid-SSBD device. Because the built−in potential of the PN junction is higher than that of the Schottky diode, the higher electric field at the edge of the active area can be evenly distributed on the guard rings to avoid breakdown of the active area.

Figure 7a,b shows how the breakdown voltage of the c-SSBD device is influenced by guard ring design. As the gap width between the innermost guard ring and active area increases, the breakdown voltage of the c-SSBD device increases. In addition, as the width of innermost guard ring increases, the breakdown voltage of the c-SSBD device decreases. Therefore, guard ring design within a proper range can increase the breakdown voltage of the c-SSBD device, such as a 10 μm gap between the active area and innermost guard ring, and a 2 μm width of the innermost guard ring. However, although a proper guard ring design can increase breakdown voltage of c-SSBD device to some extent, it remains lower than that of the Hybrid-SSBD device. In Figure 7c,d, we analyze the electric field distribution of the different designs of the c-SSBD guard ring. It can be seen that the electric field distribution at the edge of active area decreases as the gap increases, so the breakdown voltage would increase. However, increasing the innermost guard ring width would lead to increasing electric field at the edge of the active area. The higher electric field always leads to the breakdown occurring at the edge of the active area.

### 3.2. Effect of Guard Rings on Hybrid-SSBD Parameters 

The breakdown voltage of the device is sensitive to the design of the guard rings. In this section, we investigated the different guard ring designs with the aim of obtaining an optimal parameter. The simulated results are shown in Figure 8. The breakdown voltage changed with the guard ring doping depth, doping concentration, innermost ring width, and gap from the innermost ring to sensor. Figure 8a,b present the breakdown voltage as a function of guard ring doping depth and concentration. As the doping depth increases, the breakdown voltage of the device follows the trend of first increasing and then decreasing, and the optimum value for the guard ring doping depth was 1 μm. A similar pattern was observed for the function of doping concentration and breakdown voltage, but as the doping concentration exceeds 1 × 10^19^ cm^−3^, the breakdown voltage decreases slowly and tends to become saturated. We therefore consider 1 × 10^19^ cm^−3^ the optimal doping concentration to obtain a higher breakdown voltage. Figure 8c,d shows that breakdown voltage was influenced by the innermost guard ring width and the distance between the innermost guard ring (gap width) and active area. It was shown that breakdown voltage first increased and then decreased, and decreased as the gap width increased. From the simulation results, it can be concluded that the optimal innermost guard ring width and gap width were 5 μm and 3 μm, respectively.

Figure 8 shows the simulated electric field distribution at the avalanche breakdown voltage point for different guard ring designs of the Hybrid-SSBD device. The electric field distribution by guard ring doping depth is shown in Figure 9a; it can be seen that increased doping depth increasing to the expansion in the intensity of the electric field to the outer guard rings. We observed a stronger breakdown appearing the outside guard rings as the doping depth increased. However, when the doping depth is shallow, the breakdown site is still located at the edge of the active area. This is the reason why the breakdown voltage first increased and then decreased as the doping depth increased. Figure 9b suggests that the electric field intensity on the guard rings became saturated as the doping concentration increased, so when the doping concentration is greater than 1 × 10^19^ cm^−3^, the breakdown voltage exhibits a slow change. Figure 9c,d shows the electric field distribution over the different widths and gaps of the guard ring. It can be seen that the innermost guard ring width shifts the electric field outward. The electric field peak position on the guard rings reduced the electric field at the edge of the active area and avoided the breakdown caused by high electric fields. However, when the width exceeds 5 μm, the breakdown voltage would keep a small change (Figure 9c). As the gap width increased, the electric field on the guard rings would also move to the outside direction; moreover, the peak amplitude (p1p1’) increased (Figure 8d). This result would lead to easier breakdown at the edge of active area due to the increased electric field.

Based on the above observations, we believe that the guard ring structure changes the electric field distribution at the active area and guard ring edges, thereby affecting the breakdown voltage. Furthermore, the results show that the guard ring doping depth and guard ring gap greatly influence the breakdown voltage.

The simulation results of the total current density of the device with different guard ring doping depths are shown in Figure 10a. When the guard ring doping depth is shallow, the region with the highest current density is concentrated at the edge of the active area. However, when the doping depth exceeds 1 μm, breakdown occurs at the outer guard ring, so the current density at the outer guard ring increases. An increase in the guard ring gap results in stronger breakdown at the edge of the active area, and in Figure 10b, the current density increases with increasing gap width. Therefore, we can modify the breakdown voltage of the device by adjusting the structure of the guard ring, which is a good way to overcome the disadvantage of the low breakdown voltage of Schottky barrier diode detectors.

The principal source of radiation damage in semiconductor detectors is from non−ionizing energy loss (NIEL). Radiation damage introduces defects in the bulk and surface of the material that modify its behavior. In the simulation, we added three types of traps of two acceptors and one donor on the interface of silicon/SiO_2_ [20]. In order to show the difference in breakdown voltage after irradiation, the density of traps was changed from 10^11^ to 10^15^ cm^−2^. The simulation results are shown in Figure 11a,b. It can be seen that the breakdown voltage first increased and then decreased for the c-SSBD device. The breakdown voltage of the Hybrid-SSBD device also increased after radiation because irradiation generates deep level defects or banded tail states on the silicon surface. At a higher bias voltage, the surface charge will induce the opposite charge under the silicon surface. The opposite charge would generate the electric field in the opposite direction to reduce the built−in potential. Therefore, the electric field at the edge of the active area and the guard ring would reduce, increasing the breakdown voltage. However, with increasing defects, recombination would lead to increased leakage current and soft breakdown would occur in the c-SSBD device [21]. The electric field distribution of the device post−irradiation is shown in Figure 11c,d. It can be seen that the edge electric field of the c-SSBD changed with the concentration of traps. When the surface traps concentration reached 10^15^ cm^−2^, the electric field at the edge of active area was stronger, leading to a decrease in the breakdown voltage. Therefore, after the irradiation, the Hybrid-SSBD device can still maintain a higher breakdown voltage than the c-SSBD device.

Table 1 shows the comparison of surface barrier detector or radiation detector with different guard rings designed to improve the breakdown voltage. The device was reported in the references and uses an ion−implant B/P at the metal border as the edge termination [9,22,23,24,25,26]. The device can obtain a breakdown voltage of more than 500 V. The field plate structure is commonly used to increase the breakdown voltage of the radiation detector, as reported in works by Boughedda [22] and Bortoletto [24]. However, in this work, it is more convenient to improve the breakdown voltage using the hybrid structure, which can obtain a breakdown voltage of more than 1500 V.

## 4. Conclusions

In this paper, we simulate different guard ring structures for Schottky barrier diode devices. The Hybrid-SSBD device has been designed to increase the breakdown voltage of the Schottky barrier diode. The Hybrid-SSBD device uses a PN junction as a guard ring to effectively increase the electric field strength at the edge of the active area, consequently increasing the breakdown voltage of the device to more than 1500 V. Afterwards, we investigated different guard ring designs for Hybrid-SSBD devices, and found the guard ring doping depth and gap width have a critical impact on the breakdown voltage. These designs all affect the breakdown voltage of the device by changing the distribution of the fringing electric field in the active area. Therefore, for Hybrid-SSBD devices, the optimal characteristics of the guard ring are 1 × 10^19^ cm^−3^ doping concentration, 1 μm doping depth, and innermost guard ring width and gap width of 5 μm and 3 μm, respectively. We simulated the device breakdown voltage after irradiation, the Hybrid-SSBD retained a higher breakdown voltage than the c-SSBD.

## Figures and Tables

**Figure 1 micromachines-13-01811-f001:**
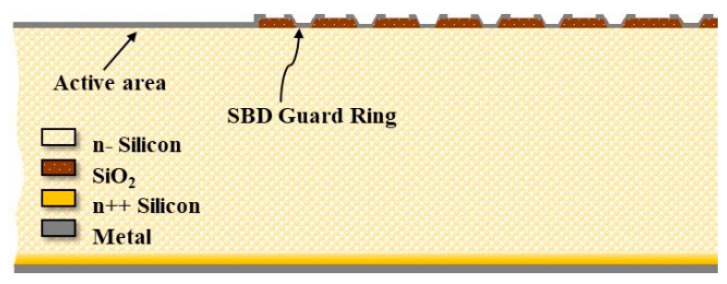
Simulation geometry, 2−D view of half−infinite and symmetric conventional SSBD structure consisting of a Schottky diode as the sensor surrounded with 7 guard rings.

**Figure 2 micromachines-13-01811-f002:**
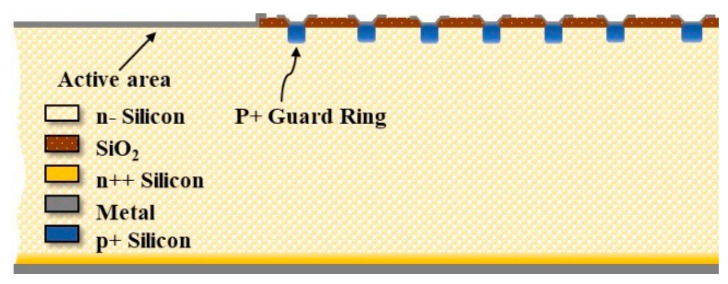
Simulation geometry, 2−D view of half−infinite and symmetric Hybrid-SSBD structure consisting of a Schottky diode as the sensor surrounded with 7 PN junction guard rings (the blue area of the p+ implantation area).

**Figure 3 micromachines-13-01811-f003:**
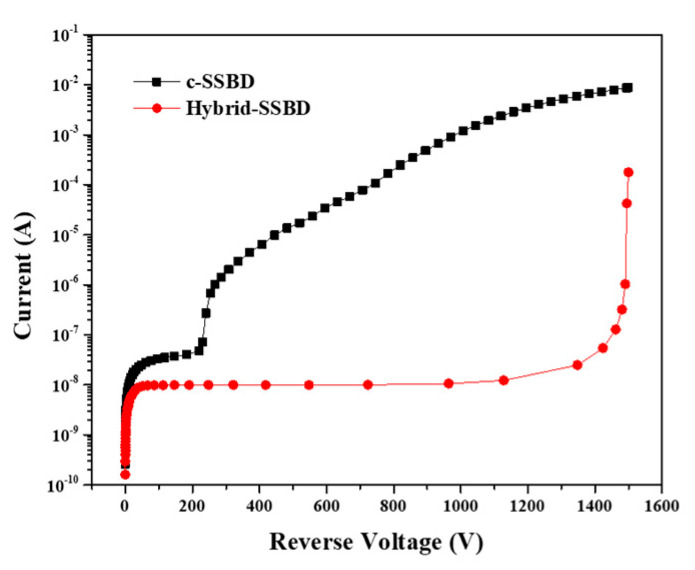
The simulated I−V curve of the two devices; the breakdown voltage is the current at rapidly rising points.

**Figure 4 micromachines-13-01811-f004:**
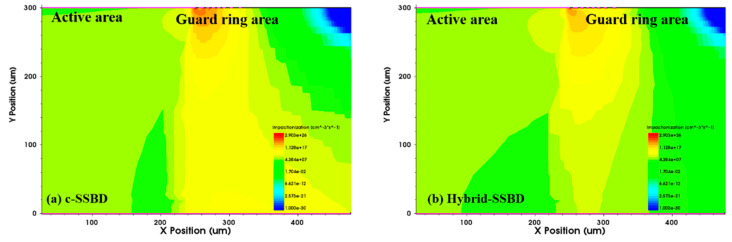
The ionizing impact distribution around the (**a**) c-SSBD and (**b**) Hybrid-SSBD devices; the colors correspond to different strengths of the impact ionization rate. The c-SSBD device had a higher impact ionization rate than the Hybrid-SSBD device at the edge of active area.

**Figure 5 micromachines-13-01811-f005:**
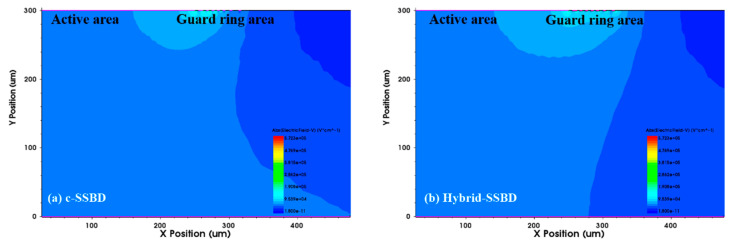
The electric field distribution of the (**a**) c-SSBD and (**b**) Hybrid-SSBD devices at a reverse bias of 1500 V. The stronger electric field is concentrated on the edge of the active area for c-SSBD, but more homogenously distributed on the guard rings for Hybrid-SSBD.

**Figure 6 micromachines-13-01811-f006:**
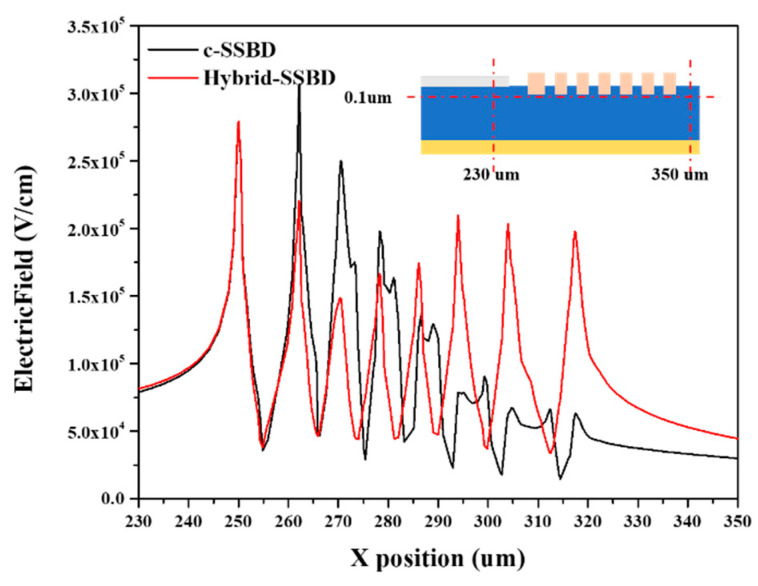
The electric field distribution at the depth of 0.1 μm under the device surface for c-SSBD and Hybrid-SSBD at a reverse bias of 1500 V. The insert graph represents the position in the device structure.

**Figure 7 micromachines-13-01811-f007:**
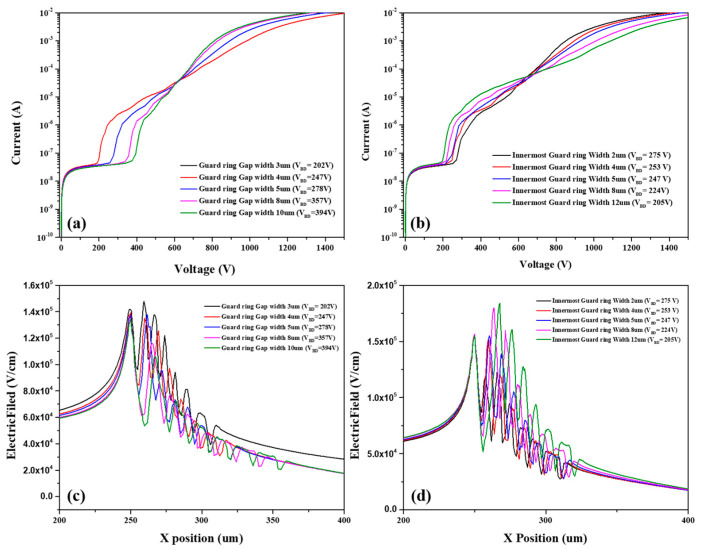
The simulated I–V curve breakdown voltage of c-SSBD is influenced by guard ring design: (**a**) spacing between the innermost guard ring and the active area, (**b**) innermost guard ring width, the electric field distribution of (**c**) spacing between the innermost guard ring and the active area, (**d**) innermost guard ring width.

**Figure 8 micromachines-13-01811-f008:**
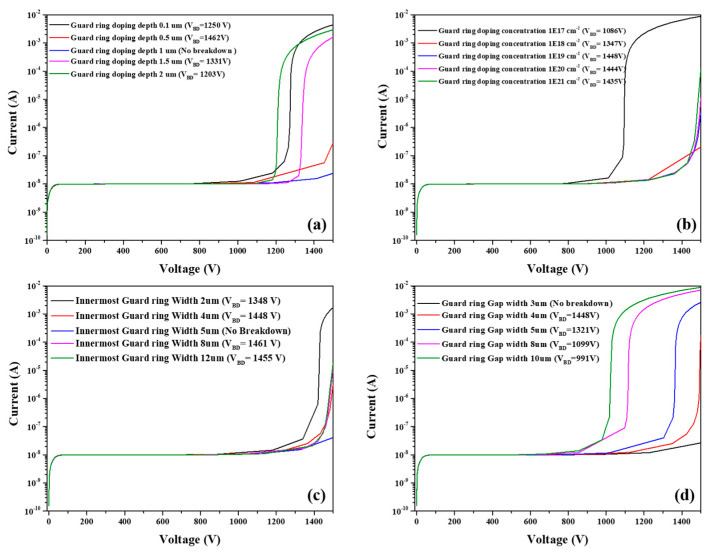
The simulation of I–V curves for the different guard ring parameters: (**a**) guard ring doping depth, (**b**) guard ring doping concentration, (**c**) innermost guard ring width, and (**d**) gap width between the active area and innermost guard ring. The optimal innermost guard ring width and gap width were 5 μm and 3 μm, respectively.

**Figure 9 micromachines-13-01811-f009:**
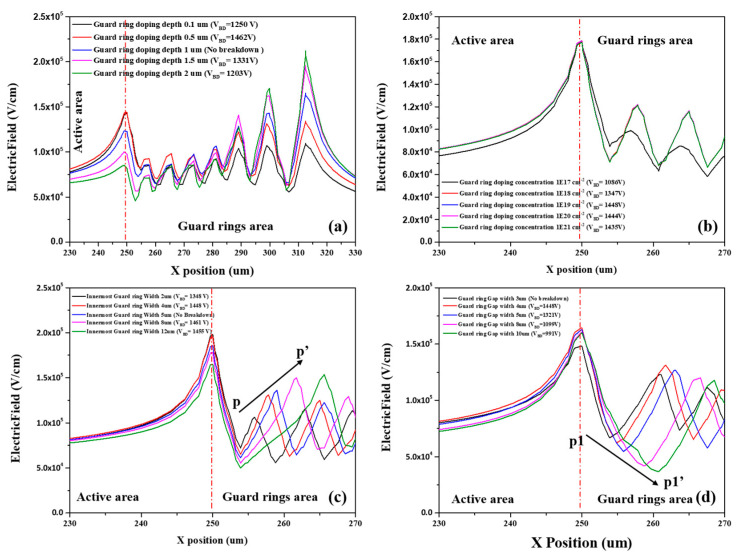
The electric field distribution at the depth of 0.1 μm under the device surface: (**a**) guard ring doping depth, (**b**) guard ring doping concentration, (**c**) innermost guard ring width, and (**d**) gap width between the active area and innermost guard ring. The red line represents the edge of the active area and pp’, p1p1’ indicates the peak amplitude change.

**Figure 10 micromachines-13-01811-f010:**
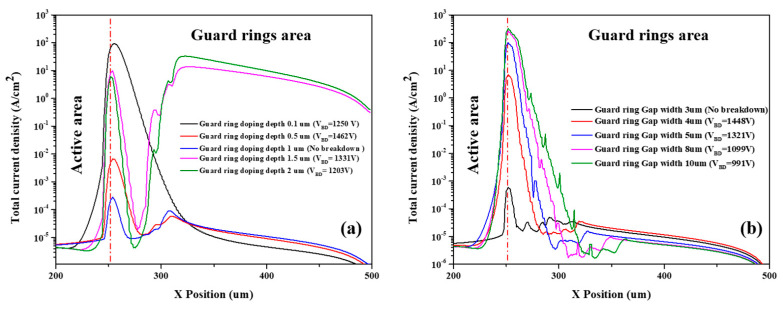
The total current density simulated results of the Hybrid-SSBD device: (**a**) guard ring doping depth, (**b**) gap width of the innermost guard ring to active area (guard ring gap width). The red line separates the two regions of the active area and guard ring area.

**Figure 11 micromachines-13-01811-f011:**
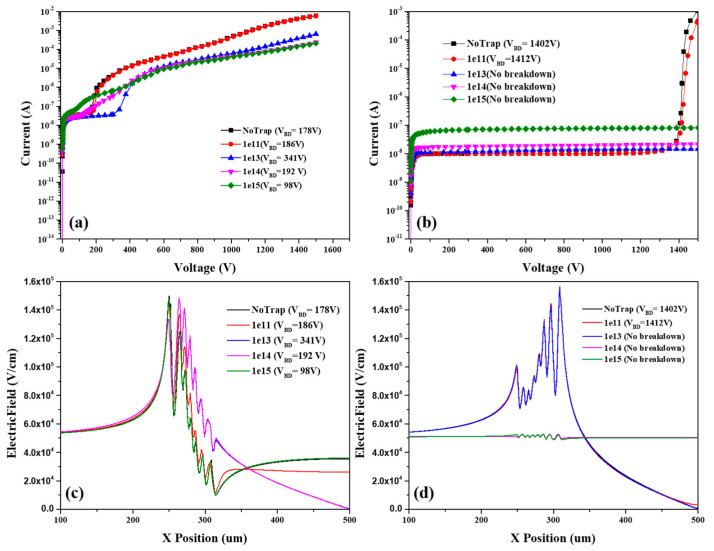
The post−irradiation performance simulation results of the I–V curve for the (**a**) c-SSBD and (**b**) Hybrid-SSBD device, and the electric field distribution of (**c**) c-SSBD and (**d**) Hybrid-SSBD.

**Table 1 micromachines-13-01811-t001:** Summary and comparison of surface barrier detector performance.

Reference	Sample Name	Substrate Resistance (Ω·cm)	Breakdown Voltage (V)
This work	c-SSBD	>10 k	<400 V
	Hybrid-SSBD		>1500 V
B.W. Liou [9]	Schottky barrier diodes	10	148
A.Boughedda [22]	Planar radiation detectors	>10 k	<700 V
M.H. Joo [23]	Au/n−Si Schottky diode	1	<400 V
D. Bortoletto [24]	Multi−guard ring p−type bulk diodes	>10 k	<1000 V
H. G. Li [25]	PtSi/Si−nanostructure detectors	20–30	100 V
F. Semendy [26]	Si−GaAs Schottky detectors	6.9 × 10^7^	<1000 V

## Data Availability

Not applicable.

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
