# Peer review of "Simulation of Silicon Surface Barrier Detector with PN Junction Guard Rings to Improve the Breakdown Voltage"

_micromachines, 2022, doi:10.3390/mi13111811_

Round 1

Reviewer 1 Report (Previous Reviewer 2)

The authors have addressed the concerns raised in the previous review in detail. The paper now highlights the fact that PN junction guard ring-based edge termination for Si Schottky barrier diodes has not been explored from an irradiation point of view. However, the general conclusions for the electric field drawn here already exist in literature. To be considered for publication, the authors should present post-irradiation performance simulation results that would evaluate the efficacy of their improved edge termination design. A benchmarking table should also be included in the manuscript, which clearly highlights the novelty and performance improvement obtained in this work w.r.t existing literature on electric fields, and post-irradiation characteristics for surface barrier detectors based on Silicon. 

Author Response

Dear reviewer

Thanks so much for these advices. The point-by-point response have been added in the word file. Please see the attachment.

Reviewer 2 Report (Previous Reviewer 1)

I appreciate the effort to improve the manuscript. However, from my point of view, the main question has not been properly answered.

For instance, the authors developed some simulations varying the spacing between inner metal ring and the active area from 3 to 10 microns. But it is expected that a relatively large spacing will not improve enough the breakdown voltage. Therefore, it is not clear the reasons to use such spacing values.

Hence, it is difficult to analyze if the manuscript is making a fair comparison between the c-SSBD and Hybrid-SSBD performance.

Author Response

Dear reviewer

Thanks so much for these advices. The point-by-point response have been added in the word file. Please see the attachment.

Round 2

Reviewer 2 Report (Previous Reviewer 1)

I appreciate the effort of the authors to improve the manuscript. From my point of view comparative Table 1 gives some evidence that Hybrid-SSBD are indeed improved over some optimization schemes of the conventional diodes.

I suggest accepting the manuscript.

Additional remark: In line 190, I guess it should be indicated Figure 9, instead of Figure 8.

This manuscript is a resubmission of an earlier submission. The following is a list of the peer review reports and author responses from that submission.

Round 1

Reviewer 1 Report

The paper is interesting, the main conclusion is the fact that the pn junction guard rings improve the diode breakdown voltage. However, there are several points to improve

1.- Figure 3 shows a comparison between the reverse current of the hybrid-SBD and the conventional one.
Please specify the guard ring width and spacing between them as well as the spacing between the main diode and the first ring.
2.- In Figure 4, please define the bias voltage considered.
3.- Are Figures 4-6 using the same devices than Figure 3?
4.- Along the paper, the proposed device has been compared with a conventional SBD diode using Schottky diodes as guard rings. However, the impact of metal rings over breakdown voltage is modified by the metal length and the spacing between metals. Therefore, a fair comparison must be done between the proposed device and the conventional SBD with a proper design to improve the breakdown voltage.

Reviewer 2 Report

The authors have presented a simulation study of a Silicon Schottky barrier diode with PN junction guard rings for efficient edge termination to improve breakdown characteristics. Although the authors have highlighted the importance of guard rings for ongoing research on wide bandgap semiconductors (GaN, Ga2O3, Diamond), they fail to comprehend the same in current Silicon technology, where power switching modules with breakdown voltages as high as 6.5 kV exists in the literature. The generic understanding of the importance of guard ring depth and its distance from the active area is not new and has already been explored in Si, SiC, and GaN technologies. For the manuscript to be published, a detailed introduction highlighting the importance of current work in the context of current/state-of-art Si Schottky devices is needed, and conclusions/benchmarking based on widely accepted figures of merits (FoMs) should be drawn. The references mentioned as contemporary research in the manuscript date back to 2000, which is not acceptable. In light of these aspects, the manuscript in its current form cannot be considered for publication.